# Diarrhea treatment center (DTC) based diarrheal disease surveillance in settlements in the wake of the mass influx of forcibly displaced Myanmar national (FDMN) in Cox's Bazar, Bangladesh, 2018

Abu S. G. Faruque[1]*, Azharul Islam Khan[1], S. M. Rafiqul Islam[1], Baitun Nahar[1], M. Nasif Hossain[1], Yulia Widiati[2], A. S. M. Mainul Hasan[2], Mukeshkumar Prajapati[3], Minjoon Kim[2], Maya Vandenent[2], Tahmeed Ahmed[1]

1 icddr,b, Dhaka, Bangladesh, 2 UNICEF Bangladesh, Cox's Bazar Field Office, Cox's Bazar, Bangladesh, 3 World Health Organization, Cox's Bazar, Bangladesh

☯ These authors contributed equally to this work.

* gfaruque@icddrb.org

**Data Availability Statement:** The data underlying this study cannot be shared publicly because of

## Abstract

### Background

In August 2017, after a large influx of forcibly displaced Myanmar nationals (FDMN) in Cox's Bazar, Bangladesh diarrhea treatment centers (DTCs) were deployed. This study aims to report the clinical, epidemiological, and laboratory characteristics of the hospitalized patients.

### Methods

The study followed cross-sectional design. In total 1792 individuals were studied. Other than data, a single, stool specimen was subjected to one step rapid visual diagnostic test for *Vibrio cholerae*. The provisionally diagnosed specimens of cholera cases were inoculated into Cary-Blair Transport Medium; then sent to the laboratory of icddr,b in Dhaka to isolate the colony as well as perform antibiotic susceptibility tests. Data were analyzed by STATA and analyses included descriptive as well as analytic methods.

### Results

Of the total 1792 admissions in 5 DTCs, 729 (41%) were from FDMN settlements; children <5 years contributed the most (n = 981; 55%). Forty percent (n = 716) were aged 15 years and above, and females were predominant (n = 453; 63%). Twenty-eight percent (n = 502) sought treatment within 24h of the onset of diarrhea. FDMN admissions within 24h were low compared to host hospitalization (n = 172, 24% vs. n = 330, 31%; p<0.001). Seventy-two percent (n = 1295) had watery diarrhea; more common among host population than FDMN (n = 802; 75% vs. n = 493; 68%; p<0.001). Forty-four percent admissions (n = 796) had some or severe dehydration, the later was common in FDMN (n = 46; 6% vs. n = 36; 3%, p =

authors are committed to maintain confidentiality of the study participants. Data are available from the icddr,b Institutional Review Board (aahmed@icddrb.org) or the corresponding author for researchers who meet the criteria for access to confidential data.

**Funding:** UNICEF, Bangladesh Grant number: GR1875 URL: https://www.unicef.org/bangladesh/en. The funders had no role in study design, data collection and analysis, decision to publish, or preparation of the manuscript.

**Competing interests:** The authors have declared that no competing interests exist.

0.005). FDMN often used public taps (n = 263; 36%), deep tube-well (n = 243; 33%), and shallow tube well (n = 188; 26%) as the source of drinking water. Nearly 96% (n = 698) of the admitted FDMN used pit latrines as opposed to 79% (n = 842) from the host community (p<0.001). FDMN children were often malnourished. None of the FDMN reported cholera.

## Conclusion

No diarrhea outbreak was detected, but preparedness for surges and response readiness are warranted in this emergency and crisis setting.

## Introduction

In August 2017, Bangladesh experienced a large sudden influx of Forcibly Displaced Myanmar Nationals (FDMN) settled in the Cox's Bazar district located in the southeast of the country. The exodus was one of the world's most recent, largest, and sudden displacement events of more than 750,000 individuals within 17 weeks [1–4], on top of already 450,000 displaced population settled in the areas, bringing the total number of displaced individuals at more than one million as of August 2019 [5–7].

To accommodate the sudden influx of such a large number of displaced people, pre-existing registered settlements were not sufficient. This resulted in the rise and development of spontaneous makeshift settlements on steep hillsides and vacant mostly low-lying areas between agriculture plots. The expansion of existing settlements caused the loss of a vast area of the adjacent forest areas [8]. In turn, this has led to environmental degradation including further erosion of land surface areas within the FDMN settlements and in the immediate surrounding areas where the host Bangladesh population is the residents [9].

Immediate critical needs escalated in the areas of shelter and non-food day-to-day commodities, food and nutrition support, safe water and optimal sanitation as well as easy availability of health care services mostly for children, women of reproductive age, and elderly individuals [5]. This influx did lead to significant pressure on the established yet fragile health system of Bangladesh, specifically in Cox's Bazar district. FDMN arrived with diverse yet often immediate health needs including gunshot wounds, burns, gender-based violence, communicable and non-communicable diseases, and needs related to mental health. [5, 6, 10–18].

A large-scale humanitarian response was almost immediately initiated, comprised of the Bangladesh Government, UN agencies, and a large number of international and national non-governmental organizations (NGOs). However, inadequate resources and supplies failed to meet the initial basic needs of the large number of displaced populations living in the settlements. Lack of adequate Water and Sanitation, Hygiene (WASH) related practices such as open defecation, use of unsafe water for domestic purposes, poor hygienic practices, overcrowding and increased mobility of the displaced population were prevailing. These threats posed immediate risks to the health of this population. Additionally, these potential risks were mounted by poorly organized public health facilities due to the lack of adequate supplies and professional health care staff [5, 14, 19–22].

Almost immediately after the large scale influx and settlement of FDMN, icddr,b, and UNICEF jointly conducted a brief field assessment in the settlements in Ukhiya and Teknaf subdistricts of Cox's Bazar. The assessment identified the potential threats of diarrheal disease outbreaks including cholera and shigellosis following which strategies were formulated to initiate mitigation measures. Accordingly, icddr,b decided to partner with UNICEF to strengthen

health care for diarrhea and associated malnutrition. The partnership included the strengthening of knowledge and skills of the health workforce in the management of acute watery diarrhea (AWD) episodes and associated malnutrition and preparedness for outbreaks of AWD. The partnership aimed to (i) train doctors, nurses, and community health workers of the government and NGO run facilities serving the FDMN in the settlements as well as host population living in the neighborhood housing; (ii) manage cases of diarrheal disease and associated malnutrition at five diarrhea treatment centers (DTC) in Leda (provided care round-the-clock), Shyamlapur (remained open round-the-clock), Balukhali (served as out-patient), Ukhiya (provided services round-the-clock), and Teknaf (remained open round-the-clock) in Cox's Bazar district; and iii) carry out DTC based diarrheal disease surveillance. The Government of Bangladesh with technical support from icddr,b partnering with international agencies and international and national NGOs undertook a massive oral cholera vaccination campaign from October 2017 to December 2018 as a pre-emptive measure to alleviate the threat of a cholera outbreak [23–25].

During the ongoing threats of cholera outbreaks [26], limited information was available among displaced and host population living in settlements and neighborhood host community on the water and sanitation (WASH) practices of the families, infant and young child feeding (IYCF) practices of children aged 0–35 months, nutritional status of children and women of childbearing age. Moreover, there was a dearth of knowledge about the common associated bacterial enteric pathogens and rotavirus in hospitalized patients in that area. Additionally, it became essential in detecting any disease outbreak immediately, particularly cholera and shigellosis so that early warning and response system can take prompt measures before any spread; thus, the control strategies are less difficult and more effective in avoiding unexpected deaths.

We aimed to report results from our ongoing diarrheal disease surveillance efforts in Ukhiya and Teknaf sub-districts of Cox's Bazar, specifically on patients who were hospitalized in the newly deployed icddr,b operated DTCs serving FDMN and the host community populations during April-December, 2018. We would also like to report on WASH practices of the families hospitalized with diarrhea; IYCF practices of those aged 0–35 months hospitalized in DTCs, nutritional status of the under-five children and women of childbearing age, and the associated common bacterial enteric pathogens and rotavirus.

## Materials and methods

### Study design

The study followed a cross-sectional facility-based surveillance design that monitored patients hospitalized most often due to dehydrating diarrheas in icddr,b operated DTCs. Patients came from both the settlements as well as neighborhood host communities in Cox's Bazar district. The majority had a duration of admission for less than 12 hours, and a small proportion was hospitalized for more than 12 hours including at least a single night stay.

### Study site

The study was conducted in Ukhiya and Teknaf sub-districts of Cox's Bazar, Bangladesh during Aril to December, 2018.

### Study population

The study population was FDMN living in the largely scattered settlements as well as Bangladesh nationals residing in the neighborhood who sought care as admissions from the icddr,b run DTCs. The host population was included in the study because they were living nearby and

sought care from the immediate catchment areas of the DTCs, they had easy access to the settlements, their mixing with the displaced population was mostly due to cultural similarities, and they shared equal threats of disease outbreaks. Moreover, from disease epidemiology perspectives, there were possibilities of any rapid spread of disease outbreaks from the neighborhood host population to the displaced vulnerable population. This study will assist in understanding several differentials when compared between displaced population and host population.

**Definitions.** Diarrhea has been defined as loose, watery stools, three or more times a day. Dysentery is an inflammatory disease of the intestine, especially of the colon resulting in diarrhea with the presence of blood and mucus in the stools.

## Sample size

To calculate sample size, we assumed the proportion of the main outcome variable displaced population hospitalized with acute watery diarrhea episodes as 40%, desired precision as 2.5%, and 5% level of significance, the minimum sample size was 1475. Considering a 10% non-response rate, the minimum sample size for this cross-sectional study design was 1639. The study had an analyzable sample size of 1792 hospitalized individuals.

## Data collection

In implementing treatment of a relatively large number of diarrhea patients with weekly monitoring, evaluation, and reporting, icddr,b followed its expertise gathered from its hospital-based Diarrheal Disease Surveillance System (DDSS) which is in operation in icddr,b's urban Dhaka and rural Matlab facilities [27]. Administering structured questionnaires, trained research assistants interviewed all hospitalized patients and/or their attendants to collect relevant information on socioeconomic and demographic profile, water and sanitation, dwelling households and surrounding environments, feeding practices of infants and young children 0–35 months old, and use of drugs and fluid therapy at home before presenting to DTCs. Standard structured questionnaires consisting of basic demographic, clinical, and associated variables of study interest (see data analysis section) were completed.

Field research assistants involved in administering field-tested questionnaires were university graduates in any discipline, with a minimum of 15 years of schooling. They had an average of 8 years' work experience in carrying out similar activities. Each morning, the supervisor checked the precision of all anthropometric instruments and calibrated the equipment. Frequent spot checks were conducted to observe the data collection process including performance in rapid diagnostic tests. Any detected error was immediately resolved.

## Stool sample collection and testing

A single, stool specimen (of at least 3 g) was collected directly from the patients following hospitalization. Immediately, one step rapid visual diagnostic test for *Vibrio cholerae* was performed by the Crystal VC dipstick test kit (161C101-05, 161C101-10, 161C101-50), ARKRAY Healthcare Pvt. Ltd., Surat, India). To facilitate microbial culture to confirm the rapid diagnostic test results; (i) the provisionally diagnosed specimens of cholera cases (the stool mostly, or swab) were inoculated into Cary-Blair Transport Medium; the medium was then sent to the clinical microbiology laboratory in Dhaka, Bangladesh as soon as possible to isolate the colony as well as to perform antibiotic susceptibility tests. Other specimens were submitted routinely once or twice a week [27].

## Statistical analysis

In this study, the differential characteristics of the hospitalized displaced and host population were explained by age, sex, nutritional status, water source etc. Study population (displaced and host population) was our outcome variable of interest. Explanatory variables included in the analysis were; demographic characteristics: age and sex; clinical features: duration of diarrhea, stool character, dehydration status, and ORS use; nutritional status: type of child nutrition, severe malnutrition of children, breastfeeding status of children 0–23 months old, and nutritional status of women 15–49 years; environmental factors: water source, and toilet use pattern; and associated common enteric pathogens: *Vibrio cholerae* O1, *Shigella*, *Salmonella*, and rotavirus. Data were analyzed by STATA (version 15.0 IC, College Station, TX: StataCorp LLC) and analyses included descriptive as well as analytic methods. Descriptive statistics were used to summarize the data, including frequencies and proportions for categorical variables. Characteristics based on the exposure were compared using Pearson Chi-square tests for categorical variables. Several statistical plots such as histograms, bar diagrams, pie charts, and scatterplots were used for data visualization. Categorical explanatory variables were coded as: age of the female respondent (15 years and more = 1, <15 years = 0), sex (female = 1, male = 0), duration of diarrhea (≥1 day = 1, <1 day = 0); dehydration status (some-severe = 1; none = 0); ORS use at home (no = 1, yes = 0); drinking water source (public tap = 1, other = 0); user of pit latrine without water seal (yes = 1, other = 0); stool character (watery = 1, other = 0); and rotavirus (yes = 1, no = 0). First, simple logistic regression analyses were undertaken for both outcome and explanatory variables to examine the association between the outcome variable and all explanatory variables separately. Variables that were observed to be significantly associated (p-value <0.05) with the outcome variable (study population) along with clinically relevant non-significant but important variables that have public health implications were included in the multivariable logistic model using stepwise backward selection method. The Hosmer–Lemeshow Goodness of fit test was run to test whether the model fitted well or not. Odds ratios (ORs) with 95% confidence intervals (95% CIs) were estimated to assess the strength of association between the outcome variable and the explanatory variables of interest. A p-value <0.05 indicated strong evidence against the null hypothesis. We also checked multicollinearity among explanatory variables using variance inflation factor (VIF). No variable with a VIF>5 was identified.

## Ethical consideration

The present study entitled: "Surveillance for etiologic agents, care-seeking behavior, the status of IYCF and WASH practices among patients or their caregivers from Rohingya refugees as well as host population in Cox's Bazar district attending icddr,b operated Diarrhea Treatment Centers" was approved by icddr,b's institutional review board (consists of Research Review Committee and Ethical Review Committee). The data collection process was understood to cause no harm including psychological distress to the participants. Data collection, measurements of nutritional status, and collection of stool specimen were carried out after obtaining voluntary informed written consent from the respondents. In case of children, the consent of their parents and/or guardians was obtained. When they were unable to read, the consent form was read aloud to the participant or his/her guardian/parents. A copy of the consent form was given to the respondent for his/her reference. The consent form was written in simple Bangla language, so that it is easily understood by the participants, even those with little or no formal schooling. In case of participants 11–17 years of age, in addition to their assent, consent of their parents and/or guardians was also obtained. The staff members clearly mentioned to the participants that answering the questions will not cause any risk to him/her or his/her

family, and there would be no direct benefit to him/her or his/her child responding to the questionnaire. Moreover, their participation in the study might serve as a groundwork for an intervention program among the FDMN and host population that would look for benefitting him/her and others in the community by implementing better health care services. Participants were informed that they or their family members would not get any remuneration for participation, and they would not have to pay any compensation for participating in the study. They were clearly informed about their rights to withdraw themselves at any point of the interview as well as the study. All precautionary measures were taken to keep participants' information confidential. The individuality of the participants was stored in locked cabinets and password-protected computer files and only key researchers had access to that information. The dataset contained the name and address of the participants, but that information was not used during the analysis, writing the report, or the manuscript.

## Results

### Care seeking at DTCs

Of the total 1792 admissions in 5 DTCs, 729 (41%) were from FDMN settlements. Children <5 years old contributed the most (n = 981; 55%) to the number of hospitalized individuals in DTCs with an identical male: female ratio (1:1). More under-5 children from the host community were hospitalized than their peers from the displaced population (59% vs. 49%; p<0,001). Overall, 40% of hospitalized patients were aged 15 years and above, more frequently from settlements than their counterparts of the same age from the host community (p = 0.002) (Table 1). Initially, care-seeking from these DTCs was low but after a well-organized communication campaign mostly with community leaders in particular, and displaced population, in general, we observed a surge like overall care-seeking round the clock in these facilities particularly during October-December 2018. Hospitalizations were also concomitantly much higher during October-December 2018 (Fig 1).

### Diarrhea, dehydration status, and use of ORS

Twenty-eight percent of the individuals hospitalized in DTCs sought care within 24 hours of the onset of clinical signs, while 7% of the admissions reported after 3 days of the onset of the diarrheal episode. Hospitalization of the displaced population within 24 hours of onset was significantly less common than their peers from the host community. However, their hospital admissions were significantly more frequent than that of the host community population in those having diarrhea for 1–3 days. Nearly 72% of the admissions presented with watery diarrhea. Such watery episodes were observed more often among the host patient population than the displaced population (75% vs. 68%; p<0.001). Eleven percent of the patients reported mucoid episodes with or without blood present in the stool. Forty-four percent of the admissions

**Table 1. Age-sex distribution of the patients attending DTCs, April-December 2018, Cox's Bazar, Bangladesh.**

| Indicator | All Population n = 1792 (%) | Displaced Population n = 729 (%) | Host Population n = 1063 (%) | P-value |
|---|---|---|---|---|
| **Age** | | | | |
| <5y | 981 (54.7) | 354 (48.6) | 627 (59.0) | <0.001 |
| % Female | 41.4 | 39.8 | 42.3 | |
| 5-14y | 95 (5.3) | 51 (7.0) | 44 (4.1) | 0.011 |
| % Female | 41.1 | 37.3 | 45.5 | |
| 15y and more | 716 (40.0) | 324 (44.4) | 392 (36.9) | 0.002 |
| % Female | 63.4 | 63.6 | 63.3 | |

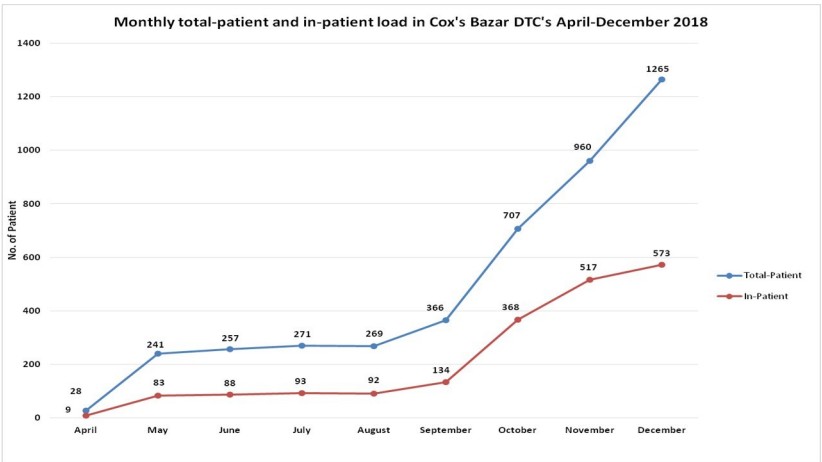

**Fig 1. Monthly total-patient and in-patient load in Cox's Bazar DTCs, April-December 2018, Cox's Bazar, Bangladesh.**

were with some form of dehydration. Severe dehydration was revealed more commonly among the FDMN than admissions from the host community (6% vs. 3%, p = 0.005). About two-thirds of the patient population used ORS at home before presenting to the DTCs. Such practice was significantly higher among the host population than FDMN (72% vs. 59%, p) (Table 2).

## IYCF practices and assessment of the nutritional status of the children aged 6–59 months and women 15–49 years' old

The proportion of 0-23-month-old babies being breastfed at the time of hospitalization was high in both communities (93% displaced young children vs. 95% host children, p = 0.63) (Table 3). Knowledge about immediate administration of colostrum to the newborn babies was significantly higher among the host population than the displaced population (70% vs. 55%; p = 0.001). Ninety-one percent of the host community population as opposed to 86% of

**Table 2. Duration of diarrhea, stool character, dehydration status, and ORS use at home before attending DTCs*, April-December 2018, Cox's Bazar, Bangladesh.**

| Indicator | All Population n = 1792 (%) | Displaced Population n = 729 (%) | Host Population n = 1063 (%) | P-value |
|---|---|---|---|---|
| **Duration of diarrhea** | | | | |
| <1 day | 502 (28.0) | 172 (23.6) | 330 (31.0) | <0.001 |
| 1–3 days | 1174 (65.5) | 517 (70.9) | 657 (61.8) | <0.001 |
| ≥4 days | 116 (6.5) | 40 (5.5) | 76 (7.1) | 0.191 |
| **Stool character** | | | | |
| Watery | 1295 (72.3) | 493 (67.6) | 802 (75.4) | <0.001 |
| Loose | 309 (17.2) | 158 (21.7) | 151 (14.2) | <0.001 |
| Mucoid±blood | 188 (10.5) | 78 (10.7) | 110 (10.3) | 0.873 |
| **Dehydration status** | | | | |
| None | 996 (55.6) | 395 (54.2) | 601 (56.5) | 0.349 |
| Some | 714 (39.8) | 288 (39.5) | 426 (40.1) | 0.847 |
| Severe | 82 (4.6) | 46 (6.3) | 36 (3.4) | 0.005 |
| **ORS use** | | | | |
| ORS use at home | 1198 (66.9) | 432 (59.3) | 766 (72.1) | <0.001 |

*DTC: Diarrhea Treatment Centers

**Table 3. Nutritional status of the children, 6–59 months old, presented with diarrhea episodes to DTCs\*, April-December 2018, Cox's Bazar, Bangladesh.**

| Indicator | All Population n = 923 (%) | Displaced Population n = 330 (%) | Host Population n = 593 (%) | P-value |
|---|---|---|---|---|
| **Children aged 6–59 months** | | | | |
| **Nutritional status** | | | | |
| Stunting | 150 (16.3) | 71 (21.5) | 79 (13.3) | 0.002 |
| Underweight | 288 (31.2) | 135 (40.9) | 153 (25.8) | <0.001 |
| Wasting | 287 (31.1) | 119 (36.1) | 168 (28.3) | 0.018 |
| **Severe malnutrition** | | | | |
| Severe Acute Malnutrition WHZ < -3 z-score | 77 (8.3) | 32 (9.7) | 45 (7.6) | 0.324 |
| Severe Acute Malnutrition MUAC < 115 mm | 29 (3.1) | 16 (4.8) | 13(2.2) | 0.043 |
| Global Acute Malnutrition MUAC < 125 mm | 192 (20.8) | 90 (27.3) | 102 (17.2) | <0.001 |
| Severe Chronic Malnutrition HAZ < -3 z-score | 45 (4.9) | 26 (7.9) | 19 (3.2) | 0.007 |
| **Breastfeeding status** | | | | |
| % Breastfed (0–23 months) | 827/979 (94.1) | 292/313 (93.3) | 535/566 (94.5) | 0.553 |
| **Women 15–49 years** | | | | |
| Height (<145.00 cm) | 64/575 (11.1) | 34/261 (13.0) | 30/314 (9.6) | 0.236 |
| BMI (<18.5) | 76/573 (13.3) | 35/261 (13.4) | 41/312 (13.1) | 0.977 |

\* DTC: Diarrhea Treatment Centers

SAM: Severe Acute Malnutrition; MUAC: mid-upper-arm circumference; WHZ: weight-for height z score; HAZ: height-for age z score

Stunting: height-for-age z score <-2

Underweight: weight-for-age z score <-2

Wasting: weight-for height z score <-2

Severe Acute Malnutrition: WHZ <-3 z-score or MUAC < 115mm

Global Acute Malnutrition: MUAC < 125 mm

Severe Chronic Malnutrition: HAZ <-3 z-score

the FDMN knew that breastfeeding should be initiated within one hour of birth of newborn babies (p = 0.024). Mostly cereal-based complementary feeds were given to children less than 2 years old and these feeds were given more often to displaced children than host children (42% vs. 34%; p = 0.011).

Forty-one percent of the hospitalized displaced under-five children were underweight (weight-for-age z-score <-2, WAZ) followed by wasted (36%; weight-for-height z-score, WHZ <-2) and stunted (22%; height-for-age z-score, HAZ <-2). Malnourishment, including severe acute malnutrition (MUAC < 115 mm) and severe stunting (HAZ <-3), were more frequent among hospitalized settlement children than the admission children from host communities (Table 3).

## WASH practices

The major sources of drinking water of the hospitalized displaced individuals were: public taps installed in the settlements (n = 263; 36%), deep tube-well (n = 243; 33%), and shallow tube well (n = 188; 26%). However, the use of deep tube well water was less common in the hospitalized settlement population than hospital admissions from the host community (n = 243, 33% vs. n = 494, 47%; p<0.001). So, was the scenario for shallow water use for hospital admissions among FDMN compared to the host population (n = 188, 26% vs. n = 443, 42%; p<0.001). Nearly 96% (n = 698) of the displaced admissions used pit latrines as opposed to 79% (n = 842) of the patients from the host community (p<0.001). However, the use of pit latrine without water seal was more common in settlements than in the host community (n = 612, 84% vs. n = 606, 57%; p<0.001). Moreover, the use of pit latrine with water seal was observed less

**Table 4. Water sources for household consumption of patients reporting to DTCs\*, April-December 2018, Cox's Bazar, Bangladesh.**

| Indicator | All Population n = 1792 (%) | Displaced Population n = 729 (%) | Host Population n = 1063 (%) | P-value |
|---|---|---|---|---|
| **Water source** | | | | |
| Public tap | 301 (16.8) | 263 (36.1) | 38 (3.6) | <0.001 |
| Deep tube well | 737 (41.1) | 243 (33.3) | 494 (46.5) | <0.001 |
| Shallow tube well | 631 (35.2) | 188 (25.8) | 443 (41.7) | <0.001 |
| Others | 123 (6.9) | 35 (4.8) | 88 (8.3) | 0.006 |
| **Toilet use pattern** | | | | |
| Pit latrine without water seal | 1218 (68.0) | 612 (84.0) | 606 (57.0) | <0.001 |
| Pit latrine with water seal | 322 (18.0) | 86 (11.8) | 236 (22.2) | <0.001 |
| Others | 252 (14.1) | 31 (4.3) | 221 (20.8) | <0.001 |

\*DTC: Diarrhea Treatment Centers

commonly among FDMN than the host community population (n = 86, 12% vs. n = 236, 22%; p < .001) (Table 4).

## Pathogen isolation

*Aeromonas* and rotavirus were the leading enteric pathogens associated with diarrheal illnesses among DTC admissions. Patients with rotavirus episodes reported more often from the host community than their peers from displaced communities (39% vs. 27%, p<0.001) (Table 5). None of the 729 stool specimens from hospitalized displaced individuals yielded the growth of *Vibrio cholerae* in their stool specimens during the study period. However, *Vibrio cholerae* O1 was detected from 5 hospitalized individuals from the host community and none of them received any oral cholera vaccine (OCV).

## Factors associated with hospitalization of the displaced population in DTCs

The association of explanatory variables after adjusting for covariates with outcome variable revealed that hospitalized FDMN were significantly more likely to report after 1 day and more (aOR 1.15, 95% CI 1.01, 1.31), drinking water from public tap (aOR 17.82, 95% CI 12.17, 26.10), user of pit latrines without water seal (aOR 4.06, 95% CI 3.10, 5.31), not user of ORS at home before coming to the DTCs (aOR 1.89, 95% CI 1.49, 2.39), and less likely to get

**Table 5. Detection of enteric pathogens in the fecal specimen of diarrhea patients presenting to DTCs\*, April—December 2018, Cox's Bazar, Bangladesh.**

| Indicator | All Population n = 1792 (%) | Displaced Population n = 729 (%) | Host Population n = 1063 (%) | P-value |
|---|---|---|---|---|
| *Vibrio cholerae* | 5 (0.3) | 0 (0.0) | 5 (0.5) | 0.162 |
| *Aeromonas* | 226 (12.6) | 92(12.5) | 134 (12.6) | 0.949 |
| Rotavirus | 611 (34.1) | 195 (26.7) | 416 (39.1) | <0.001 |
| *Vibrio cholerae* non O1/O139 | 28 (1.6) | 12 (1.6) | 16 (1.5) | 0.966 |
| Other Vibrios | 22 (1.3) | 6 (0.8) | 16 (1.5) | 0.285 |
| *Shigella* | 10 (0.6) | 3 (0.3) | 7 (0.7) | 0.714 |
| *Plesiomonas shigelloides* | 7 (0.4) | 3 (0.4) | 4 (0.4) | 0.789 |
| *Salmonella* | 58 (3.3) | 23 (3.1) | 35 (3.4) | 0.979 |
| No pathogen | 906 (50.5) | 417 (57.2) | 489 (46.0) | <0.001 |

\* DTC: Diarrhea Treatment Centers

**Table 6. Factors associated with hospitalization of displaced population from settlements in DTCs, April—December 2018, Cox's Bazar, Bangladesh.**

| | | Unadjusted OR (95% CI) | p-value | Adjusted OR (95% CI) | p-value |
|---|---|---|---|---|---|
| Duration of diarrhea 1 day and more | | | | | |
| | Yes | 1.46 (1.18–1.81) | <0.001 | 1.15 (1.01–1.31) | 0.03 |
| | < 1 day | Reference | | Reference | |
| Watery stool | | | | | |
| | Yes | 0.68 (0.55–0.84) | <0.001 | 0.76 (0.59–0.98) | 0.032 |
| | No | Reference | | Reference | |
| User of ORS at home before coming | | | | | |
| | Yes | Reference | | Reference | |
| | No | 1.77 (1.45–2.16) | <0.001 | 1.89 (1.49–2.39) | <0.001 |
| Drinking water from public tap | | | | | |
| | Yes | 15.22 (10.65–21.76) | <0.001 | 17.82 (12.17–26.10) | <0.001 |
| | No | Reference | | Reference | |
| User of pit latrine without water seal | | | | | |
| | Yes | 3.94 (3.13–4.97) | <0.001 | 4.06 (3.10–5.31) | <0.001 |
| | No | Reference | | Reference | |
| Rotavirus | | | | | |
| | Yes | 0.57 (0.46–0.69) | <0.001 | 0.59 (0.46–0.75) | <0.001 |
| | No | Reference | | Reference | |
| Age 15y and more | | | | | |
| | Yes | 1.36 (1.13–1.65) | 0.001 | | |
| | < 15 y | Reference | | | |
| Moderate-severe dehydration | | | | | |
| | Yes | 1.1 (0.91–1.33) | 0.348 | | |
| | No | Reference | | | |
| Female sex | | | | | |
| | Yes | 0.99 (0.83–1.20) | 0.985 | | |
| | No | Reference | | | |

admission with rotavirus diarrhea (aOR 0.59, 95% CI 0.46, 0.75) and watery stool (aOR 0.76, 95% CI 0.59, 0.98) (Table 6).

## Discussion

In this study, we have explored the health, nutrition, and socio-demographic status of the DTC admissions among displaced as well as the host population living in settlements and neighborhood host communities respectively. The findings of this study have public health importance and may be useful for the health system of Bangladesh for preparedness in situations with risks from public health perspectives such as cause-specific diarrhea surges in both the host and displaced population.

Several findings related to care-seeking from DTCs were noteworthy. Females aged 15 years and more living in settlements and host communities were more often hospitalized for diarrheal episodes than their male peers.

Vulnerability of pregnant and lactating women and school-aged girls to increased rates of diarrheal diseases including cholera has been repeatedly indicated by several studies. Researchers have postulated that severely malnourished women are likely to have compromised immune systems due to diminished nutritional supply that often fails to meet their needs in emergency and crisis settings, thus they are becoming more prone to diarrheal diseases

including cholera. Women and school going girls are more exposed to higher infective doses of causative agents because of their active involvement in domestic activities such as taking care of sick family members, washing their dirty clothes, and cleaning toilets. Additionally, they are involved in fetching water, handling untreated water and raw food particularly during cooking which could ultimately become contaminated. Thus, their existing immunity is at risk to be superseded by the infective dose of an enhanced number of causative organisms [28–33].

ORS use at home was significantly lower in the displaced population than those living in the host community. Likely explanations are fewer accesses to ORS packets at the household or community level because of less organized outreach activities to promote ORS at the household level. Alternatively, the displaced population was lacking awareness as well as knowledge about ORS use before coming to DTCs as well as early care seeking. Strengthening of prevailing outreach activities can address these issues. Use of hygiene toilets like pit latrine with water seal among the displaced population needs more promotion. Access to more safe water (chlorinated water supplied through taps installed at community level) was observed in settlements mostly for FDMN as provided by international agencies and NGOs. However, their access to deep and shallow tube well water was less compared to that of hospital admissions from the host community. Hospitalizations less likely due to associated rotavirus infection and watery stool may be a result of proper management of these illnesses at the household level with ORS and appropriate feeding during the episode.

Similar to other tropical countries, *Aeromonas* spp. was the most commonly isolated bacterial pathogen among the diarrhea admissions in DTCs. *Aeromonas* spp. is known to be widespread in freshwater, estuarine, and marine environment. They can even survive at most extreme salt concentrations in coastal and seawater. *Aeromonas* spp. is associated with self-limiting acute watery diarrhea (a few may even present with "rice-water" stools), often with vomiting or dysenteric diarrhea [34]. In the absence of cholera cases, admissions were less likely due to rotavirus diarrhea and watery stool.

In emergencies, displaced children are well known to be at higher risk of widespread under-nutrition and micronutrient deficiencies as adequate food and health services are often not readily available. A cross-sectional study conducted in an internally displaced population in northern Uganda reported a high prevalence of stunting among camp children. The study indicated that children, 3–24 months old were at higher risk of acute malnutrition [35]. It has been observed in emergency and crisis settings that those who were malnourished before the beginning of the crisis, their vulnerability further gets enhanced during the crisis. Moreover, researches have indicated that children become undernourished much earlier in emergency and crisis settings than adults. The frail immune system in acutely malnourished children makes them at higher risk for infectious diseases, they need more time to recover, and the disease may even result often in a fatal outcome. A cross-sectional study was undertaken among children aged 6–23 (n = 236) months who were residents of Teknaf and Ukhia sub-districts in 2014 reported early initiation of breastfeeding in 44% of infants [36]. Ten percent of the hospitalized under-five FDMN children and 8% from the host community had severe acute malnutrition (WHZ <-3 z-score). Such rates were above the Bangladesh national statistics (4–5%). Moreover, global acute malnutrition (GAM; MUAC <125 mm) was 27% which was far above the critical emergency threshold (15%) in children living in settlements. In Bangladesh, about 24% of women suffer from short stature (height <145 cm), such short stature was observed among 13% of the displaced women as opposed to 10% of women from the host community. These women are at higher risk for several adverse health and nutritional outcomes [37–39].

## Implications of the findings for global public health understanding

The upsurge of diseases is likely to occur in the wake of humanitarian emergencies if the susceptibility of the population is compromised due to physical and mental stress, shortage of adequate food, and lack of basic needs of life including water, sanitation, and hygiene (WASH). Children among the displaced population were also at risk of infectious disease outbreaks because of the presence of malnourished children with depressed immunity that may result in upsurges of childhood infectious diseases like measles, gastroenteritis, and respiratory infections.

It is well known that the absence of WASH measures are threats for diarrhea, cholera and infectious disease outbreaks and such threats may continue to persist till the situation is not improved [40, 41]. Outbreaks of measles and diphtheria among the children dwelling in settlements have been reported soon after their arrivals [10, 42–45].

The provision of improved water supply in fairly adequate quantity and monitoring of its quality started soon after the arrival of FDMN. Because of the extreme needs, shallow tubewells were poorly installed at the beginning. Although that led to immediate access to water, its quality was questionable. Soon many shallow tube-wells became non-functional. Public health concerns emerged when FDMN were observed to bathe, wash, and practice open defecation in surface water sources in their neighborhood. They were noticed to drill holes for the collection of underground water; however, the water quality from those sources was uncertain from a basic safe drinking water point of view. Fecal contamination of drinking water was high and water quality assessments in the camps revealed that 92% of the water specimens were contaminated with *Escherichia coli* while half (48%) of the specimens were observed to be exceedingly contaminated (>100 CFU/100 ml). The quality of drinking water was a major concern as 50% of the water samples at sources (collection point) and 89% of the samples at the household level (usage point) were reported to be contaminated during the routine water quality judgment and monitoring activities [3, 46, 47].

The sanitation issues were key challenges in settlements because of overcrowding and lack of enough dwelling spaces. Latrines were shared in settlements and the basic public health standards were not maintained properly. To immediately meet the excess needs of the largely arrived settlement population, maintenance of conventional minimum depth of pit latrines were ignored and they were installed often in close to each other. Moreover, routine desludging became a major awkward issue due to an excess number of FDMN in settlements. Access to soap was limited to 50% of the displaced families that deferred them from adopting appropriate hygiene practices like hand washing at the household level, making them highly vulnerable to public health risks of diarrhea, cholera, and dysentery outbreaks [46–50].

In 1978, Bangladesh observed the arrival of the displaced population from Myanmar for the first time. Published data from the clinics operated for them reported that 28% of visits were due to watery diarrhea, 32% were dysentery, and 40% cases presented with other illnesses. Of 2321 diarrhea stool specimens subjected to microbial testing by culture, 29% yielded growth of pathogens. Among these enteric pathogens, 22% were *Shigella* alone, and 6% detected to be *V. cholerae*. Coliform counts of water were extremely high. Deaths were common among infants, children, and elderly individuals. These deaths were mostly due to acute watery diarrhea (12%), fever (23%), and consequent poor nutrition (52%) [41]. In 2015, acute watery diarrhea (AWD) accounted for 7–9% of morbidities in the camps, and of all consultations that they sought, 22% were due to AWD [51]. The sentinel surveillance for cholera patients conducted by the icddr,b, and Institute of Epidemiology and Disease Control Research (IEDCR) of the Government of Bangladesh in the district hospital of Cox's Bazar since 2014 reported that like other parts of Bangladesh cholera outbreaks also occurred among the population of Cox's

Bazar district. Thus traditionally, Bangladesh's host population of Ukhiya and Teknaf continued to be at risk of cholera. The sentinel surveillance further indicated concurrent hospitalization of cholera cases in Cox's Bazar district hospital when the exodus (August-September 2017) of FDMN was extremely high in Ukhia and Teknaf sub-districts [23]. Currently, the absence of any outbreaks of cholera or shigellosis may be a good reflection of mass cholera vaccine campaigns as well as improved WASH facilities in settlements.

Both the host and displaced populations are currently at risk of acute infectious diseases including outbreaks of water-borne diseases. The displaced population is being closely monitored by international agencies through early warning, and alert response system (EWARS) and emergency surveillance systems [10]. These agencies are ensuring uninterrupted water quality surveillance in settlements and households. However, there is absence of strong routine programs that aim at the improvement of personal and kitchen hygiene of the displaced population. Thus, more WASH improvements and management practices of diarrheal disease cases are needed. International agencies and NGOs, as well as local NGOs run diarrhea treatment facilities, are performing rapid diagnostic tests (RDT) supplied readily by WHO-Cox's Bazar for detection of *V. cholerae* in stool specimens of patients from settlements as well as the host community for early detection of cholera outbreaks. As soon as the presence of *V. cholerae* is detected provisionally; the stool specimen is collected by icddr,b surveillance team, and dispatched to the central laboratory of icddr,b in Dhaka for confirmation by conventional culture method following standard laboratory procedures. According to DTC based surveillance, displaced children are often malnourished and thus they are also at higher risk for severe dehydrating cholera episodes [52–54]. Women living in high-risk areas, mostly disadvantaged communities are vulnerable to rapidly progressing dehydrating diarrheal episodes including cholera. Moreover, these women are at risk of dying and if they are pregnant, that may even lead to adverse pregnancy outcomes due to severe dehydrating diarrheal episodes. Because of the high fecundity of displaced women, a large number of pregnant women particularly adolescents with pregnancy are at a continued higher risk of cholera and its associated life-threatening complications [55–58]. Continuation of vigilance for cholera in settlements supports the Government of Bangladesh's health system activities in the sub-districts. Vigilance for cholera is linked to the application of rapid diagnostic tests and laboratory culture methods for the detection of *V. cholerae* infections [59–65]. Such endeavor is important from the public health point of view in detecting disease outbreaks quickly before any disease spreads to take place which is expected to keep the disease burden at a minimal level and save more lives [10].

## Limitations of the study

One of the limitations of the study was that these activities were DTC based and only those getting hospitalized in DTCs have been included in the study. Patients with less severe disease who reported to the facility and received care as the out-patient basis and those cases that occurred at the community and did not report to DTCs have not been studied. Thus, results may not be generalizable.

## Strengths of the study

A large number of patients and quality laboratory performance were the strengths of the study. The study has highlighted the importance of DTC-based diarrheal disease surveillance with round-the-clock capturing of patients with a timely laboratory back-up and immediate reporting which can enhance the preparedness and response team in early detection of outbreaks followed by the institution of preventive and control measures in emergency and crisis settings.

Moreover, DTC based surveillance facilitated direct comparisons between the displaced population and the host community population concerning a good number of key variables.

## Conclusion

With effective DTC networking, laboratory-based disease surveillance, and OCV campaigns in the present emergency and crisis settings, no cholera or acute watery diarrhea outbreak was observed during the study period. Preparedness for surges as well as response readiness is warranted in this emergency and crisis settings. Threats of cholera outbreaks among the settlement population are continuing due to new arrivals of the FDMN with compromised host susceptibility, the declining immunity of OCV among the OCV recipients as well as an increasing number of cohort children without any exposure to OCV. Preparedness for surges and vigilance for cholera cases should be the priority undertakings because of existing threats of cholera in both the host and displaced population.

## Acknowledgments

We acknowledge the contribution of icddr,b's core donors including the Government of the People's Republic of Bangladesh, Global Affairs Canada, Canada; Swedish International Development Cooperation Agency and the Foreign, Commonwealth and Development Office (Department for International Development previously), UK Aid for their continuous support and commitment to the icddr,b's research efforts.

## Author Contributions

**Conceptualization:** Abu S. G. Faruque, Azharul Islam Khan, Tahmeed Ahmed.

**Data curation:** Abu S. G. Faruque.

**Investigation:** Abu S. G. Faruque, Azharul Islam Khan, Tahmeed Ahmed.

**Methodology:** Abu S. G. Faruque, Azharul Islam Khan, S. M. Rafiqul Islam, Baitun Nahar, M. Nasif Hossain.

**Project administration:** S. M. Rafiqul Islam, Baitun Nahar, M. Nasif Hossain.

**Supervision:** M. Nasif Hossain.

**Writing – original draft:** Abu S. G. Faruque, Azharul Islam Khan, S. M. Rafiqul Islam, Baitun Nahar, M. Nasif Hossain.

**Writing – review & editing:** Abu S. G. Faruque, Azharul Islam Khan, S. M. Rafiqul Islam, Baitun Nahar, M. Nasif Hossain, Yulia Widiati, A. S. M. Mainul Hasan, Mukeshkumar Prajapati, Minjoon Kim, Maya Vandenent, Tahmeed Ahmed.

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
