## [Decision Letter · Decision Letter 0]

4 Jan 2021

PONE-D-20-28913

Diarrhea Treatment Centre (DTC) based diarrheal disease surveillance in settlements in the wake of the mass influx of Forcibly Displaced Myanmar National (FDMN) in Cox’s Bazar, Bangladesh, 2018.

PLOS ONE

Dear Dr. Faruque,

Thank you for submitting your manuscript to PLOS ONE. After careful consideration, we feel that it has merit but does not fully meet PLOS ONE’s publication criteria as it currently stands. Therefore, we invite you to submit a revised version of the manuscript that addresses the points raised during the review process.

We look forward to receiving your revised manuscript.

Kind regards,

Ivan D. Florez

Academic Editor

PLOS ONE

Journal Requirements:

5. We note that you have included the phrase “data not presented” in your manuscript. Unfortunately, this does not meet our data sharing requirements. PLOS does not permit references to inaccessible data. We require that authors provide all relevant data within the paper, Supporting Information files, or in an acceptable, public repository. Please add a citation to support this phrase or upload the data that corresponds with these findings to a stable repository (such as Figshare or Dryad) and provide and URLs, DOIs, or accession numbers that may be used to access these data. Or, if the data are not a core part of the research being presented in your study, we ask that you remove the phrase that refers to these data.

Additional Editor Comments:

Your manuscript has been reviewed by two experts in the field, and they have found some points that need to be addressed before this manuscript is considered for publication. Please go through the reviewers' comments and consider addressing these points, and prepare a revised version.

Reviewers' comments:

Reviewer's Responses to Questions

**Comments to the Author**

1. Is the manuscript technically sound, and do the data support the conclusions?

Reviewer #1: Partly

Reviewer #2: Yes

2. Has the statistical analysis been performed appropriately and rigorously? 

Reviewer #1: No

Reviewer #2: Yes

3. Have the authors made all data underlying the findings in their manuscript fully available?

Reviewer #1: No

Reviewer #2: No

4. Is the manuscript presented in an intelligible fashion and written in standard English?

Reviewer #1: Yes

Reviewer #2: Yes

5. Review Comments to the Author

Reviewer #1: Abstract:

Study objectives should be clearly mentioned.

Materials and methods section should be more informative. For instance, the section should include information on the study design, study population, sampling technique, study tool and data analysis plan.

Introduction:

Background information is nicely organized

Justification should be stronger.

Study objective is not clear. It should be clearly mentioned.

Materials and methods:

Study design should come first, before study site and so on.

Why Bangladesh nationals were included as the study population? Including Bangladesh nationals might dilute the study findings.

Line 140-150: Unnecessary information.

Sample size:

o Why the prevalence of enteric pathogen (0.3) was used for estimating sample size? Is this the main outcome variable?

Data collection:

o How was the quality of the collected data ensured?

Statistical Analysis:

o Please clearly state what are the outcome and explanatory variables.

o Only descriptive statistics seems unsatisfactory for publishing in a renowned journal. Some sorts of inferential statistics should be performed (linear/logistic regression etc.).

Ethical consideration:

o Please mention whether informed written consent was obtained or not?

o What about anonymity and confidentiality?

Results:

In the method section, it is stated that the study performed only descriptive analysis, however, in the result section “P-value” has been reported. Please explain how was the “P-value” obtained.

Formatting of the Tables should be improved by following those in international journals.

Discussion:

Discussion section should be enriched by explaining the study findings using existing literature. For instance, authors stated “This may be due to the increased vulnerability of females to diarrheal illnesses because of their higher compromised immunity or excess exposure to contaminated water and food during household activities.” (Line 312-314). This explanation should be backed up by relevant reference.

Please mention the major strengths of the study.

Adding a “Conclusion” section would be beneficial.

Reviewer #2: GENERAL COMMENTS

General editing for grammar and flow of the content

It is not clear who the refugees are, where they immigrated from?

Check on style of in-text citation especially where more than one article is citted. The numbers are separated with [] instead of commas.

INTRODUCTION

Line 66 – what is FDMN, write in full the first time. This goes for any abbreviation in the text.

Lines 91-94- the authors have listed examples of basic services that were not adequate. Can the authors give examples of amounts of these services that were available for refugees? How did they determine that these services were not adequate?

Lines 100-103- what were some specific findings from these initial assessments conducted by icddr,b, and UNICEF at the two mentioned sub-districts of Cox Bazar?

Line 110-113- can the authors clarify to the readers if these locations: Leda (operated round-the-clock),

Shyamlapur (round-the-clock), Balukhali (served as out-patient), Ukhiya (round-the-clock), and

Teknaf (round-the-clock) in Cox’s Bazar district were camps where the were displaced populations were resettled or they were names of clinics?

MATERIALS AND METHODS:

STUDY POPULATIONS - were the settlements for displaced populations separate from where local populations were living or these two populations were integrated?

What were the study objectives and hypothesis?

DATA COLLECTION- were theses face to face interviews or they were a mixture of interviews and record reviews? For example how was the nutritional data collected? Please clarify how the different types of data was collected.

RESULTS

Generally, the authors have presented results on various variables by comparing the situations of displaced populations with local populations at the two study locations. However, what is not clear is how much of the basic services e.g. water, sanitation, food, health services were available to the two populations respectively. What types of food were available and how much? Were there feeding centers for the malnourished children? How much water on average was available per person per day? How were these services operated in the settlements? What were the shortfalls? Were the sphere standards met? When the authors assert that certain services were not adequate, what do they mean? The context of response by the government and relief agencies to the displaced and local populations need to be described more clearly. This will help with interpreting the findings and putting them within the context. Also the authors need to suggest some practical concrete recommendations. This can only be possible if they explain the current response situation and provision of services.

6. PLOS authors have the option to publish the peer review history of their article (what does this mean?). If published, this will include your full peer review and any attached files.

Reviewer #1: No

Reviewer #2: No

---

## [Author Response · Author response to Decision Letter 0]

14 Feb 2021

Comment(s)

 Response

Abstract:

Study objectives should be clearly mentioned.

Materials and methods section should be more informative. For instance, the section should include information on the study design, study population, sampling technique, study tool and data analysis plan. 

 Authors express sincere gratitude to the reviewer for sharing valuable comments and suggestions. All authors recognize that comments and suggestions are very generous and important to strengthen the manuscript. The revised objectives now read: The present study aimed to report the characteristics of the hospitalized patients in the newly deployed DTCs, focusing additionally on water and sanitation practices of the families, infant and young child feeding practices, nutritional status of the under-five children and women of childbearing age, and the associated common bacterial enteric pathogens and rotavirus.

Please see page 1 in the revised version.

The materials and methods section read: The study was conducted in Ukhiya and Teknaf sub-districts of Cox’s Bazar, Bangladesh, and followed a cross-sectional design. The study population was forcibly displaced Myanmar nationals (FDMN) living in the largely scattered settlements as well as Bangladesh nationals residing in the neighbourhood who sought care at the icddr,b run DTCs, and enrolled into the facility based diarrheal disease surveillance. In total 1792 hospitalized individuals were considered as study participants. Other than relevant data, a single, stool specimen was collected, immediately one step rapid visual diagnostic test for Vibrio cholerae was performed by the Crystal VC dipstick test kit. The provisionally diagnosed specimens of cholera cases were inoculated into Cary-Blair Transport Medium; then sent to the Clinical Microbiology Laboratory of icddr,b in Dhaka, Bangladesh as soon as possible to isolate the colony as well as perform antibiotic susceptibility tests. Data were analysed by STATA (version 15.0 IC, College Station, TX: StataCorp LLC) and analyses included descriptive as well as for analytic methods. 

Please see page 1 and 2 in the revised version.

Introduction:

Background information is nicely organized

Justification should be stronger. 

Study objective is not clear. It should be clearly mentioned. As per suggestion, necessary additions and revisions have been made. Please see page 3 line 58 in the revised justification section now reads: During the ongoing threats of cholera outbreaks [26], limited information was available among displaced and host population living in camps and neighbourhood host community on the water and sanitation (WASH) practices of the families, infant and young child feeding (IYCF) practices of children aged 0-35 months, nutritional status of children and women of childbearing age. Moreover, there was a dearth of knowledge about the common associated bacterial enteric pathogens and rotavirus in hospitalized patients in that area. Additionally, it became essential in detecting any disease outbreak immediately, particularly cholera and shigellosis so that early warning and response system can take prompt measures before any spread; thus, the control strategies are less difficult, and more effective in avoiding unexpected deaths. 

Please see page 7 and 8 in the revised version.

The revised study objective now reads: We aimed to report results from our ongoing diarrheal disease surveillance efforts in Ukhiya and Teknaf sub-districts of Cox’s Bazar, specifically on patients who were hospitalized in the newly deployed icddr,b operated DTCs serving FDMN and the host community populations during April-December, 2018. We would also like to report on WASH practices of the families hospitalized with diarrhoea; IYCF practices of those aged 0-35 months hospitalized in DTCs, nutritional status of the under-five children and women of childbearing age, and the associated common bacterial enteric pathogens and rotavirus.

Please see page 7 and 8 in the revised version.

Materials and methods:

Study design should come first, before study site and so on.

Why Bangladesh nationals were included as the study population? Including Bangladesh nationals might dilute the study findings.

Line 140-150: Unnecessary information. 

 Thank you for spotting this. 

All required revisions have been made now. 

Now the study design is placed before the study site and so on. Please see page 3 line 59-60 in the revised version. 

The host population was included in the study because they were living nearby and sought care from the immediate catchment areas of the DTCs, they had easy access to the settlements, their mixing with the displaced population was mostly due to cultural similarities, and they shared equal threats of disease outbreaks. Moreover, from disease epidemiology perspectives, there were possibilities of any rapid spread of disease outbreaks from the neighbourhood host population to the displaced vulnerable population. The study will assist in understanding several differentials when compared between displaced population and host population. 

Please see page 8 and 9 in the revised version.

Thank you very much for spotting this. All necessary deletions have also been made. 

Sample size:

Why the prevalence of enteric pathogen (0.3) was used for estimating sample size? Is this the main outcome variable?

 Gross revisions have been undertaken as per comments and suggestions. Please see page 3 line 59-60 in the revised version which now reads:

To calculate sample size, we assumed the proportion of the main outcome variable the females aged 15 years and above hospitalised with acute watery diarrhea episodes which was 63%, desired precision as 3%, and 5% level of significance, the minimum sample size was 995. Considering 5% non-response rate the total sample size of the cross-sectional study design was ~1050.

 Please see page 9 in the revised version.

Data collection:

How was the quality of the collected data ensured?

 Thank you for raising this issue We have added a statement under data collection section in page 4 lines 99-100 that reads: Field research assistants involved in administering field-tested questionnaires were university graduates in any discipline, with minimum of 15 years of schooling. They had an average of 8 years’ work experience in carrying out similar activities. Each morning, the supervisor checked the precision of all anthropometric instruments and calibrated the equipments. Frequent spot checks were conducted to observe data collection process including performance in rapid diagnostic tests. Any detected error was immediately resolved.

Please see page 10 in the revised version.

Statistical Analysis:

Please clearly state what are the outcome and explanatory variables.

Only descriptive statistics seems unsatisfactory for publishing in a renowned journal. Some sorts of inferential statistics should be performed (linear/logistic regression etc.). Authors thank to reviewer for important insight. The paragraph has been revised to incorporate analysis details. We agree with the reviewer’s concern. Please see the revised version page 6 lines 143-149 now reads: Study population (displaced and host population) was our outcome variable of interest. Explanatory variables included in the analysis were; demographic characteristics: age and sex; clinical features: duration of diarrhea, stool character, dehydration status, and ORS use; nutritional status: type of child nutrition, severe malnutrition of children, breastfeeding status of children 0-23 months old, and nutritional status of women 15-49 years; environmental factors: water source, and type of toilet use pattern; and associated common enteric pathogens: Vibrio cholerae O1, Shigella, Salmonella, and rotavirus.

Data were analysed by STATA (version 15.0 IC, College Station, TX: StataCorp LLC) and analyses included descriptive as well as for analytic methods. Descriptive statistics were used to summarize the data, including frequencies and proportions for categorical variables. Characteristics based on the exposure were compared using Pearson X2 tests for categorical variables. Several statistical plots such as histograms, bar diagrams, pie charts and scatterplots were used for data visualization. 

Categorical explanatory variables were coded as: age of the female respondent (15 years and more=1, <15 years=0), sex (female=1, male=0), duration of diarrhea (≥1 day=1, <1 day=0); dehydration status (some-severe=1; none=0); ORS use at home (no=1, yes=0); drinking water source (public tap=1, other=0); user of pit latrine without water seal (yes=1, other=0); stool character (watery=1, other=0); and rotavirus (yes=1, no=0). 

All explanatory variables were analysed initially in a univariate model, and the attributes that were observed to be significantly associated (p-value <0.05) with the outcome variable (study population) were considered to be included in the multivariable logistic regression model. 

Finally, logistic regression was used to assess the relationship between the outcome variable and explanatory variables with a binary outcome. First, simple logistic regression analyses were undertaken for both outcome and explanatory variables to examine the association between the outcome variable and all explanatory variables separately. Variables that were observed to be significantly associated (p-value <0.25) with the outcome variable (study population) along with clinically relevant non-significant but important variables that have public health implications were included in the multivariable logistic model using backward stepwise selection method. A Goodness of fit test was run to test whether the model fitted well or not. Odds ratios (ORs) with 95% confidence intervals (95% CIs) were estimated to assess the strength of association between outcome variable and the explanatory variables of interest, and p-value<0.05 was considered as the level of significance. We also checked multicollinearity among explanatory variables using variance inflation factor (VIF). No variable with a VIF> 5 was identified.

Please see page 10, 11 and 12 in the revised version.

Ethical consideration:

Please mention whether informed written consent was obtained or not? 

What about anonymity and confidentiality? 

 We have revised the paragraph on ethical consideration which now reads: The present study entitled: “Surveillance for etiologic agents, care-seeking behavior, the status of IYCF and WASH practices among patients or their caregivers from Rohingya refugees as well as host population in Cox’s Bazar district attending icddr,b operated Diarrhea Treatment Centers” was approved by icddr,b’s institutional review board (consists of Research Review Committee and Ethical Review Committee). The data collection process was understood to cause no harm including psychological distress to the participants. Data collection, measurements of nutritional status and collection of stool specimen were carried out after obtaining informed written consent from the respondents. In case of children, the consent of their parents and/or guardians were obtained. If they were unable to read, the consent form was read aloud to the participants or their guardians/parents. A copy of the consent form was given to the respondents for their reference. The consent form was written in simple Bangla language, so that it is easily understood by the participants, even those with little or no formal schooling. Participants who were 11-17 years of age, in addition to their assent, consent of their parents and/or guardians were also obtained. The enumerators made it clear to the participant that answering the questions will not cause any risk to him/her or his/her family, and there would be no direct benefit to him/her or his/her child responding to the questionnaire. Moreover, their participation in the study might serve as a groundwork for an intervention programme among the displaced and host population that would try to benefit him/her and others by delivering improved health care services. Participants were informed that no money would be provided to them or their family members for participation, and they would not have to pay any remuneration for participating in the study. They were also informed about their rights to withdraw themselves at any point of the interview as well as from the study. The precaution was taken to keep information about the participants confidential. Identity of the participants was stored in locked cabinets and in password-protected computer files and was available to the key researchers only. 

The dataset contained name and address of the participants, but those were not used during the analysis, writing of the report or the manuscript. 

Please see page 12 in the revised version.

Results:

In the method section, it is stated that the study performed only descriptive analysis, however, in the result section “P-value” has been reported. Please explain how was the “P-value” obtained.

Formatting of the Tables should be improved by following those in international journals.

 Thank you for spotting this error. Please see the revised version which reads: Variables with a p<0.05 in the final regression model was considered as the level of significance. 

Please see page 16-19 in the revised version.

Discussion:

Discussion section should be enriched by explaining the study findings using existing literature. For instance, authors stated “This may be due to the increased vulnerability of females to diarrheal illnesses because of their higher compromised immunity or excess exposure to contaminated water and food during household activities.” (Line 312-314). This explanation should be backed up by relevant reference.

 Vulnerability of pregnant and lactating women and school-aged girls to increased rates of diarrheal diseases including cholera has been repeatedly indicated by several studies. Researchers have postulated that severely malnourished women are likely to have compromised immune systems due to diminished nutritional supply that often fails to meet their needs in emergency and crisis settings, thus they are becoming more prone to diarrheal diseases including cholera. Women and school going girls are more exposed to higher infective doses of causative agents because of their active involvement in domestic activities such as taking care of sick family members, washing their dirty clothes, and cleaning toilets. Additionally, they are involved in fetching water, handling untreated water and raw food particularly during cooking which could ultimately become contaminated. Thus, their existing immunity is at risk to be superseded by the infective dose of an enhanced number of causative organisms [31–36].

Please see page 20 in the revised version.

Please mention the major strengths of the study.

 The major strengths of the study now read: A large number of patients and quality laboratory performance were the strengths of the study. The study has highlighted the importance of DTC-based diarrheal disease surveillance with round-the-clock capturing of patients with a timely laboratory back-up and immediate reporting which can enhance the preparedness and response team in early detection of outbreaks followed by the institution of preventive and control measures in emergency and crisis settings. Moreover, DTC based surveillance facilitated direct comparisons between the displaced population and the host community population concerning a good number of key variables. 

Please see page 26 in the revised version.

Adding a “Conclusion” section would be beneficial.

 We agree with the reviewer’s suggestion. The statement related to conclusion stands: With effective DTC networking, laboratory- based diseased surveillance, and OCV campaigns in the present emergency and crisis settings, no cholera or acute watery diarrhea outbreak was observed during the study period. Preparedness for surges is warranted in this emergency and crisis settings. 

Please see page 26 in the revised version.

---

## [Decision Letter · Decision Letter 1]

16 Apr 2021

PONE-D-20-28913R1

Diarrhea Treatment Centre (DTC) based diarrheal disease surveillance in settlements in the wake of the mass influx of Forcibly Displaced Myanmar National (FDMN) in Cox’s Bazar, Bangladesh, 2018.

PLOS ONE

Dear Dr. Faruque,

Thank you for submitting your manuscript to PLOS ONE. After careful consideration, we feel that it has merit but does not fully meet PLOS ONE’s publication criteria as it currently stands. Therefore, we invite you to submit a revised version of the manuscript that addresses the points raised during the review process.

The manuscript represents original research, that is novel and does not appear to have been published elsewhere.  The article is presented in an intelligible fashion and is written well.   However, as described in the review below the outline of the research, description of the methods, statistics and analyses performed do not meet current standards for publication at this time.

We look forward to receiving your revised manuscript.

Kind regards,

Mark Simonds Riddle

Academic Editor

PLOS ONE

Reviewers' comments:

Reviewer's Responses to Questions

**Comments to the Author**

1. If the authors have adequately addressed your comments raised in a previous round of review and you feel that this manuscript is now acceptable for publication, you may indicate that here to bypass the “Comments to the Author” section, enter your conflict of interest statement in the “Confidential to Editor” section, and submit your "Accept" recommendation.

Reviewer #1: (No Response)

2. Is the manuscript technically sound, and do the data support the conclusions?

Reviewer #1: No

3. Has the statistical analysis been performed appropriately and rigorously? 

Reviewer #1: No

4. Have the authors made all data underlying the findings in their manuscript fully available?

Reviewer #1: No

5. Is the manuscript presented in an intelligible fashion and written in standard English?

Reviewer #1: Yes

6. Review Comments to the Author

Reviewer #1: Reviewer’s Comment:

Article number: PONE-D-20-28913-R1

Title of the Article:

Diarrhea Treatment Centre (DTC) based diarrheal disease surveillance in settlements in the wake of the mass influx of Forcibly Displaced Myanmar National (FDMN) in Cox’s Bazar, Bangladesh, 2018.

Overall:

The manuscript reported some important and interesting findings and used data from a satisfactorily designed study. However, the way the manuscript was written mostly seems with a report and not a journal article. A journal article usually identifies 2-4 specific research questions and keep focus to answer those questions throughout the document. In case of the current manuscript, that consistency and coherence is not properly maintained and the study objective is too broad. I would, therefore, suggest the authors to kindly identify 1-2 important research questions (e.g., nutritional status of women and children and their associated factors, WASH practice and its associated factors etc.) and keep focus on those aspects in the revised manuscript.

Abstract:

In the abstract, it is mentioned that “In total 1792 hospitalized individuals were considered as study participants”, however, in the main manuscript, sample size was estimated as 1050 (L 174). Kindly explain the discrepancy.

Study objective:

A journal article usually identifies 2-3 specific study objectives and keep focus to those throughout the manuscript. In case of the current manuscript, the study objective is too broad (to report the characteristics of the hospitalized patients in the newly deployed DTCs, focusing additionally on water and sanitation practices of the families, infant and young child feeding practices, nutritional status of the under-five children and women of childbearing age, and the associated common bacterial enteric pathogens and rotavirus). Moreover, consistency and coherence among study objective, method (specially outcome explanatory variables) and result is not properly maintained. I would, therefore, suggest the authors to kindly identify important key topics (e.g., nutritional status of women and children and their associated factors, WASH practice and its associated factors etc.) and keep focus on those aspects in the revised manuscript.

Materials and methods

Sample size:

Authors considered the females aged 15 years and above hospitalized with acute watery diarrhea as the main outcome variable for sample size calculation (L 170-171), however, in the Statistical Analysis section (L 2014), they stated “Study population (displaced and host population) was our outcome variable”. Please explain this discrepancy.

Statistical Analysis

L 204-210: Authors stated “Study population (displaced and host population) was our outcome variable of interest. Explanatory variables included in the analysis were; demographic characteristics: age and sex; clinical features: duration of diarrhea, stool character, dehydration status, and ORS use; nutritional status: type of child nutrition, severe malnutrition of children, breastfeeding status of children 0-23 months old, and nutritional status of women 15-49 years; environmental factors: water source, and type of toilet use pattern; and associated common enteric pathogens: Vibrio cholerae O1, Shigella, Salmonella, and rotavirus”. This means in this study, the displaced and host population (outcome variable) was explained by (or depends on) age, sex, nutritional status, water source etc., which is logically not possible. Most importantly, the outcome variable should be any characteristic of study population and study population itself cannot be an outcome variable. Authors are kindly requested to clearly think about the outcome variable and explanatory variables and then perform analysis accordingly.

L 220: Authors stated that "All explanatory variables were analyzed initially in a univariate model, and the attributes that were observed to be significantly associated (p-value <0.05) with the outcome variable (study population) were considered to be included in the multivariable logistic regression model". Why did the authors perform univariate analysis for explanatory variables but not for outcome variable given that for further analysis, univariate analysis of outcome variable is also important?

L 230: Authors mentioned “A Goodness of fit test was run to test whether the model fitted well or not”. Which Goodness of fit test was used?

L 233: The statement “p-value<0.05 was considered as the level of significance” is not correct. Level of significance (expressed as alpha or α) is the probability of rejecting the null hypothesis when it is true, whereas, p-value is the probability that an observed difference could have occurred just by random chance. P-value and α are inter-related, however, a p-value cannot be considered as the level of significance.

Results

L 316: The finding presented in the Table 3 indicates Nutritional status of children is the outcome variable and its distribution is shown across the type of population (all, displaced and host), which does not match with the outcome variable stated in the statistical analysis section (L 204). Please clearly identify the outcome variable of this paper.

L 356: The sub-heading “Factors associated with hospitalization of the displaced population in DTCs” suggests that authors intended to identify which factors affected hospitalization of the displaced population, however, Table 6 indicates that they tried to compare WASH indicators (and NOT factors of hospitalization) between displaced and host population. On the other hand, the description under the sub-heading (e.g., hospitalized FDMN were significantly less likely to pass watery stool etc.) implies that the authors intended to identify difference in WASH indicators (such as watery vs. bloody diarrhoea) among hospitalized FDMN. Authors are kindly requested to thoroughly revise Table 6 and related description in the text to ensure alignment. Also, for multivariate regression tables, please report categories of covariates including reference category (e.g., variable: gender; category: male vs female, reference category: male). Please follow other articles published in Plos One for table formatting.

Best wishes.

7. PLOS authors have the option to publish the peer review history of their article (what does this mean?). If published, this will include your full peer review and any attached files.

Reviewer #1: No

---

## [Author Response · Author response to Decision Letter 1]

30 May 2021

Reviewer #1: Reviewer’s Comment:

Article number: PONE-D-20-28913-R1

Title of the Article:

Diarrhea Treatment Centre (DTC) based diarrheal disease surveillance in settlements in the wake of the mass influx of Forcibly Displaced Myanmar National (FDMN) in Cox’s Bazar, Bangladesh, 2018.

Overall:

The manuscript reported some important and interesting findings and used data from a satisfactorily designed study. However, the way the manuscript was written mostly seems with a report and not a journal article. A journal article usually identifies 2-4 specific research questions and keep focus to answer those questions throughout the document. In case of the current manuscript, that consistency and coherence is not properly maintained and the study objective is too broad. I would, therefore, suggest the authors to kindly identify 1-2 important research questions (e.g., nutritional status of women and children and their associated factors, WASH practice and its associated factors etc.) and keep focus on those aspects in the revised manuscript.

Response: Thank you very much for your valuable comment. As the threats of diarrheal disease outbreaks including cholera and shigellosis were prevailing in the settlements, therefore for better preparedness and mitigation of threats a network of DTCs was established mainly to serve the displaced population living in the settlements. Given the fact that hospitalization of the displaced population from the settlements was lower than that of host population, therefore we decided to address the research question of such a relatively low care seeking practices and assist in formulation of improved health services strategy by determining factors that were associated with hospitalization in DTCs of FDMN from settlements.

Table 6. Factors associated with hospitalization of displaced population from settlements in DTCs, April - December 2018, Cox’s Bazar, Bangladesh

Abstract:

In the abstract, it is mentioned that “In total 1792 hospitalized individuals were considered as study participants”, however, in the main manuscript, sample size was estimated as 1050 (L 174). Kindly explain the discrepancy.

Response: Thank you very much for your kind observation and asking us to explain the discrepancy. We have revised that section to explain the discrepancy as much as possible by us. Thus, to calculate sample size, we assumed the proportion of the main outcome variable the displaced population hospitalized with acute watery diarrhea episodes as 40%, desired precision as 2.5%, 5% level of significance, the minimum sample size was 1475. Considering a 10% non-response rate the minimum sample size for this cross-sectional study design was 1639. The study had an analyzable sample size of 1792 hospitalized individuals. Line 168-172

Study objective:

A journal article usually identifies 2-3 specific study objectives and keep focus to those throughout the manuscript. In case of the current manuscript, the study objective is too broad (to report the characteristics of the hospitalized patients in the newly deployed DTCs, focusing additionally on water and sanitation practices of the families, infant and young child feeding practices, nutritional status of the under-five children and women of childbearing age, and the associated common bacterial enteric pathogens and rotavirus). Moreover, consistency and coherence among study objective, method (specially outcome explanatory variables) and result is not properly maintained. I would, therefore, suggest the authors to kindly identify important key topics (e.g., nutritional status of women and children and their associated factors, WASH practice and its associated factors etc.) and keep focus on those aspects in the revised manuscript.

Response: Thank you for your kind suggestions. We have made necessary revisions by adding a few sentences to the text of the manuscript. To maintain consistency and coherence among study objective, method, and result we aimed to report results from our ongoing diarrheal disease surveillance efforts in Ukhiya and Teknaf sub-districts of Cox’s Bazar, specifically on patients who were hospitalized in the newly deployed icddr,b operated DTCs serving FDMN and the host community populations during April-December, 2018. We would also like to report on WASH practices of the families hospitalized with diarrhea; IYCF practices of those aged 0-35 months hospitalized in DTCs, nutritional status of the under-five children and women of childbearing age, and the associated common bacterial enteric pathogens and rotavirus. Additionally, we aimed to highlight the factors that were associated with the hospitalization of displaced population in the DTCs. Line 135-141.

Materials and methods

Sample size:

Authors considered the females aged 15 years and above hospitalized with acute watery diarrhea as the main outcome variable for sample size calculation (L 170-171), however, in the Statistical Analysis section (L 2014), they stated “Study population (displaced and host population) was our outcome variable”. Please explain this discrepancy.

Response: Thank you very much for your valuable suggestion. To calculate sample size, we assumed the proportion of the main outcome variable displaced population hospitalized with acute watery diarrhea episodes as 40%, desired precision as 2.5%, 5% level of significance, the minimum sample size was 1475. Considering a 10% non-response rate the minimum sample size for this cross-sectional study design was 1639. The study had an analyzable sample size of 1792 hospitalized individuals. Line 168-172.

Statistical Analysis

L 204-210: Authors stated “Study population (displaced and host population) was our outcome variable of interest. Explanatory variables included in the analysis were; demographic characteristics: age and sex; clinical features: duration of diarrhea, stool character, dehydration status, and ORS use; nutritional status: type of child nutrition, severe malnutrition of children, breastfeeding status of children 0-23 months old, and nutritional status of women 15-49 years; environmental factors: water source, and type of toilet use pattern; and associated common enteric pathogens: Vibrio cholerae O1, Shigella, Salmonella, and rotavirus”. This means in this study, the displaced and host population (outcome variable) was explained by (or depends on) age, sex, nutritional status, water source etc., which is logically not possible. Most importantly, the outcome variable should be any characteristic of study population and study population itself cannot be an outcome variable. Authors are kindly requested to clearly think about the outcome variable and explanatory variables and then perform analysis accordingly.

Response: Thanking once again for the valuable observations. Necessary revisions have been made in the manuscript accordingly. In this study, the differential characteristics of the hospitalized displaced and host population (outcome variable) were explained by age, sex, nutritional status, water source etc. The outcome variable was hospitalization of study population. Line206-207.

L 220: Authors stated that "All explanatory variables were analyzed initially in a univariate model, and the attributes that were observed to be significantly associated (p-value <0.05) with the outcome variable (study population) were considered to be included in the multivariable logistic regression model". Why did the authors perform univariate analysis for explanatory variables but not for outcome variable given that for further analysis, univariate analysis of outcome variable is also important?

Response: Our sincere thanks for sharing your kind opinion. We performed univariate analysis for outcome variable. Those texts that were duplicated were removed and we made necessary revisions. Line 217-218.

L 230: Authors mentioned “A Goodness of fit test was run to test whether the model fitted well or not”. Which Goodness of fit test was used?

Response: Thanks a lot for your kind suggestion and asking us to address your query. We have responded by adding the name of the Goodness of fit test. The Hosmer–Lemeshow test, a statistical test for goodness of fit was used for logistic regression models in this analysis. Line 229-230.

L 233: The statement “p-value<0.05 was considered as the level of significance” is not correct. Level of significance (expressed as alpha or α) is the probability of rejecting the null hypothesis when it is true, whereas, p-value is the probability that an observed difference could have occurred just by random chance. P-value and α are inter-related, however, a p-value cannot be considered as the level of significance.

Response: Many thanks for sharing your concerns. We have made necessary revisions in the text accordingly. A p-value <0.05 indicated strong evidence against the null hypothesis. Line 230. 

Results

L 316: The finding presented in the Table 3 indicates Nutritional status of children is the outcome variable and its distribution is shown across the type of population (all, displaced and host), which does not match with the outcome variable stated in the statistical analysis section (L 204). Please clearly identify the outcome variable of this paper.

Response: Thanks once again for your kind suggestions. The outcome variable was the hospitalized study population. Line 206-207.

L 356: The sub-heading “Factors associated with hospitalization of the displaced population in DTCs” suggests that authors intended to identify which factors affected hospitalization of the displaced population, however, Table 6 indicates that they tried to compare WASH indicators (and NOT factors of hospitalization) between displaced and host population. On the other hand, the description under the sub-heading (e.g., hospitalized FDMN were significantly less likely to pass watery stool etc.) implies that the authors intended to identify difference in WASH indicators (such as watery vs. bloody diarrhoea) among hospitalized FDMN. Authors are kindly requested to thoroughly revise Table 6 and related description in the text to ensure alignment. Also, for multivariate regression tables, please report categories of covariates including reference category (e.g., variable: gender; category: male vs female, reference category: male). Please follow other articles published in Plos One for table formatting.

Response: While thanking for your kind valuable suggestions we would like to mention that accordingly necessary revisions have been made in the manuscript as well as in Tables of the manuscript. The association of explanatory variables after adjusting for covariates with outcome variable revealed that hospitalized FDMN were significantly more likely to report after 1 day and more (aOR 1.15, 95% CI 1.01, 1.31), drinking water from public tap (aOR 17.82, 95% CI 12.17, 26.10), user of pit latrines without water seal (aOR 4.06, 95% CI 3.10, 5.31), not user of ORS at home before coming to the DTCs (aOR 1.89, 95% CI 1.49, 2.39), and less likely to get admission with rotavirus diarrhea (aOR 0.59, 95% CI 0.46, 0.75) and watery stool (aOR 0.76, 95% CI 0.59, 0.98) (Table 6). Necessary revisions in Table 6 have been made. Line 376-378.

---

## [Editor Report · Decision Letter 2]

28 Jun 2021

Diarrhea Treatment Centre (DTC) based diarrheal disease surveillance in settlements in the wake of the mass influx of Forcibly Displaced Myanmar National (FDMN) in Cox’s Bazar, Bangladesh, 2018.

PONE-D-20-28913R2

Dear Dr. Faruque,

We’re pleased to inform you that your manuscript has been judged scientifically suitable for publication and will be formally accepted for publication once it meets all outstanding technical requirements.

Kind regards,

Mark Simonds Riddle

Academic Editor

PLOS ONE
---

## [Editor Report · Acceptance letter]

21 Jul 2021

PONE-D-20-28913R2 

Diarrhea Treatment Centre (DTC) based diarrheal disease surveillance in settlements in the wake of the mass influx of Forcibly Displaced Myanmar National (FDMN) in Cox’s Bazar, Bangladesh, 2018. 

Dear Dr. Faruque:

I'm pleased to inform you that your manuscript has been deemed suitable for publication in PLOS ONE. Congratulations! Your manuscript is now with our production department. 

Kind regards, 

on behalf of

Dr. Mark Simonds Riddle 

Academic Editor

PLOS ONE